# A Comparison of Different Wave Modelling Techniques in An Open-Source Hydrodynamic Framework

**Weizhi Wang** [1],*,[†] , **Arun Kamath** [1] , **Tobias Martin** [1] , **Csaba Pákozdi** [2] and **Hans Bihs** [1]

1 Department of Civil and Environmental Engineering, Norwegian University of Science and Technology, 7034 Trondheim, Norway; arun.kamath@ntnu.no (A.K.); tobias.martin@ntnu.no (T.M.); hans.bihs@ntnu.no (H.B.)
2 SINTEF Ocean, 7052 Trondheim, Norway; csaba.pakozdi@sintef.no
* Correspondence: weizhi.wang@ntnu.no; Tel.: +47-96699762
† Current address: Høgskoleringen 7A, 7491 Trondheim, Norway.

**Abstract:** Modern design for marine and coastal activities places increasing focus on numerical simulations. Several numerical wave models have been developed in the past few decades with various techniques and assumptions. Those numerical models have their own advantages and disadvantages. The proper choice of the most useful numerical tool depends on the understanding of the validity and limitations of each model. In the past years, REEF3D has been developed into an open-source hydrodynamic numerical toolbox that consists of several modules based on the Navier–Stokes equations, the shallow water equations and the fully nonlinear potential theory. All modules share a common numerical basis which consists of rectilinear grids with an immersed boundary method, high-order finite differences and high-performance computing capabilities. The numerical wave tank of REEF3D utilises a relaxation method to generate waves at the inlet and dissipate them at the numerical beach. In combination with the choice of the numerical grid and discretisation methods, high accuracy and stability can be achieved for the calculation of free surface wave propagation and transformation. The comparison among those models provide an objective overview of the different wave modelling techniques in terms of their numerical performance as well as validity. The performance of the different modules is validated and compared using several benchmark cases. They range from simple propagations of regular waves to three-dimensional wave breaking over a changing bathymetry. The diversity of the test cases help with an educated choice of wave models for different scenarios.

**Keywords:** numerical wave models; high-performance computing; open-source; CFD; Navier–Stokes equations; shallow water equations; potential flow theory

## 1. Introduction

Each fluid flow is subject to the conservation laws of mass, momentum and energy which can be described by several nonlinear partial differential equations. Numerical modelling is the method of solving these equations numerically by replacing them with a set of algebraic equations. Today, this powerful technique is used in all industries and research areas, such as aero- and hydrodynamics, weather predictions or mixing processes. In contrast to experiments, numerical simulations are in general cheaper, faster in the preparation and more flexible with respect to specific external conditions or changing geometries.

Free surface flows frequently arise in nature and present an increasingly important subject due to increased sea transport, population growth and changing climate. The correct simulation of the

interfaces separating the different fluids is key knowledge in marine and hydraulic engineering. The class of interface phenomena range from current to large-scale waves of varying amplitude to splashing with coalescence and breakup situations. This variety of effects reveals the development of capable numerical models for two-phase flow applications as a difficult task.

The open-source hydrodynamics framework REEF3D [1] was originally developed to overcome these difficulties by taking the specific challenges in hydraulics, coastal and marine engineering into consideration. This affected the design choices for the grid architecture, the discretization methods of the governing equations, the treatment of the complex free surface and the computational efficiency.

The ever increasing computational resources allow the computation of more and more complex flow problems at a reasonable cost, even for small companies and research institutions. The limiting factor of such simulations becomes less the necessary computational power but rather the time it takes for the engineer to generate the numerical grids and post-process the results. However, these high-performance computations are only possible if the code provides a consistent parallelisation strategy. From the beginning, REEF3D was designed under consideration of high-performance computations (HPC). Therefore, all parts of the code are fully parallelised based on the domain decomposition strategy and the Message Passing Interface (MPI).

The numerical grid affects the range of applicability of numerical methods but also the productivity in usage. REEF3D utilises a rectilinear grid to overcome the limitations from complicated grid generation processes. In each principal direction, user-specified analytical stretching functions enable the refinement of the grid at selected locations. Ray tracing and inverse distance algorithms are included to incorporate natural bathymetries and complicated structures using the STL file format. Together with the directional immersed boundary method of Berthelsen and Faltinsen [2], this effectively simplifies the user input in pre-processing.

Suitable boundary conditions for the application in hydraulics, coastal and marine engineering have to be given. This particularly includes establishing a numerical wave tank with varies wave generation and dissipation methodologies. The level set method is used for capturing the propagation of the free surface [3]. The challenge arising from most interface models relates to physical discontinuities of the fluid properties at the interface. Low-order discretization techniques lead to a large amount of numerical diffusion, whereas high-order methods produce oscillatory and non-physical results. In order to keep a high numerical accuracy and stability, the implementation of a high-order weighted essentially non-oscillatory (WENO) scheme is the key step towards the accurate representation of sharp interfaces. The Cartesian grid makes it possible to employ the fifth-order accurate WENO scheme of Jiang and Shu [4] for all convection terms in REEF3D. Also for the discretization in time, a high-order method is selected with the third-order total variation diminishing (TVD) Runge–Kutta scheme [5]. The equations of fluid motion are solved on a staggered grid which ensures tight velocity–pressure coupling and avoids unphysical high air velocity above waves. As a result, wave propagation and transformation can be calculated throughout the wave steepness range up to the point of wave breaking and beyond, with no artificially high air velocities impacting the quality of the free surface. In the past, multiple applications proved the validity of this approach for wave propagation and wave–structure interaction. In [6], the wave generation and absorption were validated and compared to other CFD codes. Bihs et al. [7] analysed the generation, propagation and impact of wave packets using REEF3D. Breaking waves and their interaction with a complex jacket structure were investigated by Aggarwal et al. [8]. Multi-directional irregular waves were subject of the studies in [9]. Alternative approaches for a numerical wave tank based on CFD were presented in e.g., [10,11]. Both utilise a volume of fluid method with interface-compression [12] to capture the free surface and a collocated unstructured grid with second-order accuracy for the spatial and temporal discretization. The models were applied to experiments for wave propagation, and all results indicated the applicability of CFD for these kinds of problems [13–15].

The source code of REEF3D is available at http://www.reef3d.com and is published under the GPL license, version 3. REEF3D is written in an object-oriented C++ structure which enables

a module-based design. This led to the development of several extensions of the main code. For applications near the coast and in rivers, a dynamic sediment transport model and porous structures were incorporated. The simulated flow field is coupled with the morphological module in REEF3D to simulate, e.g., the scouring process around piles [16]. The morphological evolution of the sediment bed is based on the Exner formula, a modified calculation of the critical bed shear stress and a sand slide algorithm. The porous medium module solves the volume-averaged Navier–Stokes equations by adding appropriate terms and coefficients to the common Reynolds-averaged Navier–Stokes equations solved in REEF3D::CFD [17]. The model is also adapted for vegetation [18]. In Bihs and Kamath [19], a floating algorithm was presented which utilises the same directional immersed boundary method developed for fixed structures. Recently, a mooring model based on finite elements [20] was added which improves the capabilities of the model for the simulation of moored-floating structures in waves [21].

The phase-resolved modelling of the far-field is important for providing a realistic wave boundary condition for near-field CFD wave modelling. REEF3D, with its distinct numerical basis of high-order finite differences on rectilinear grids, is capable of incorporating simplified phase-resolving wave models for these types of problems.

For very large scale wave modelling, such as the wave transformation from the ocean to the coast, spectral wave models such as SWAN [22] are applicable. SWAN solves the wave action or energy balance equation, which describes the wave spectrum evolution in space and time. The model lacks the ability to resolve phases which is necessary information for more detailed analyses. Here, depth-averaged shallow water models have been favoured for the coastal and harbour wave modelling because most coastal areas share relatively shallow water conditions. Shallow water models are essentially two-dimensional and, thus, require fewer computational resources. One possible approach is based on the Boussinesq equations [23] which can accurately model wave reflection and diffraction as well as non-dispersive linear wave propagation. Extended versions of the Boussinesq equations enable the prediction of wave propagation and transformation from deep to shallow water using improved dispersive terms [24]. In contrast, REEF3D::SFLOW was introduced as a novel non-hydrostatic shallow water model following the quadratic pressure profile assumption. It benefits from the high-order discretization schemes and good scaling properties of REEF3D. Thus, large-scale coastal wave propagations over natural topography are possible.

The specific characteristic of Norwegian fjords and the general demand for fast far-field solutions in marine engineering require an alternative approach due to the changing dispersion relation in deep water regions. A potential flow solver is ideal for the fast calculation of wave propagation in deep water conditions as viscous effects are not important in the far-field domain. The general potential problem for waves is described by the Laplace equation with boundary conditions for the free surface and the bottom. This system of equations is highly nonlinear and describes a one-phase three-dimensional flow field. High-order spectral (HOS) methods [25], which solve the fully nonlinear potential problem in deep water, have gained popularity [26]. HOS methods are capable of capturing nonlinear wave interaction at a reasonable computational cost, though they are dependent on empirical input for wind forcing and wave breaking. Amongst others, Seiffert and Ducrozet [27] incorporated a wave breaking parameter in HOS-NWT [28] and simulated irregular breaking waves in 2D without wind or current. They could successfully compare surface elevation, wave spectra and energy dissipation with experiments. An alternative approach is the fully nonlinear potential flow (FNPF) model, which is based on the solution of the potential problem in physical space and time. The direct numerical solution of the Laplace equation using the method of finite differences is the basis of the model OceanWave3D [29]. This model has been used to simulate wave–structure interaction [14,30] and nonlinear wave propagation over large spatial scales with variable bathymetry [31]. The effects of wave steepness, water depth, white-capping, and directional spreading can be included with few assumptions to obtain a better description of the real sea state to calculate extreme wave statistics and wave crest height distributions. Within the REEF3D framework, REEF3D::FNPF combines the



approach of solving the Laplace equation on a $\sigma$-coordinate system using high-order finite difference methods with its high-performance computing capabilities and natural bathymetry handling.

Previously, different wave models are developed by different developers and institutes, often with various numerical implementations, making a direct comparison among the modelling techniques difficult. Now, REEF3D has evolved into an open-source numerical framework that includes several types of numerical wave modelling: a computational fluid dynamic (CFD) solver REEF3D::CFD solving the Navier–Stokes equations, a shallow water model REEF3D::SFLOW solving the non-hydrostatic shallow water equations and a fully nonlinear potential flow solver REEF3D::FNPF solving the Laplace equation with the fully nonlinear boundary conditions. With such a numerical framework, an objective comparison of the different wave modelling techniques is made possible. The authors attempt to reveal the differences in the three numerical wave modelling methods in terms of their numerical performance and physical validity by explaining the development and numerical implementations of REEF3D and testing its three modules through a series of benchmark cases.

The structure of the manuscript is arranged as the following. First, in Section 2, the development and numerical implementation of the REEF3D numerical framework and its three wave modelling modules are introduced. Then, an objective comparison among the different types of wave modules is performed using the three REEF3D wave modelling modules through a series of benchmark testings in Section 3. In the process, the evidence of the models' strengths and limitations are revealed and explained. Finally, the findings and recommendations for an educated choice of the wave models are summarised in Section 4.

## 2. Numerical Fluid Modules

### 2.1. REEF3D::CFD

Mass and momentum are conserved for an incompressible fluid by solving the continuity and Reynolds-averaged Navier–Stokes (RANS) equations

$$\frac{\partial u_i}{\partial x_i} = 0, \tag{1}$$

$$\frac{\partial u_i}{\partial t} + u_j \frac{\partial u_i}{\partial x_j} = -\frac{1}{\rho}\frac{\partial p}{\partial x_i} + \frac{\partial}{\partial x_j}\left[(\nu + \nu_t)\left(\frac{\partial u_i}{\partial x_j} + \frac{\partial u_j}{\partial x_i}\right)\right] + g_i, \tag{2}$$

with $u_i$ the velocity vector, $\rho$ the fluid density, $p$ the pressure, $\nu$ and $\nu_t$ the kinematic and turbulent viscosity, and $g_i$ the gravity acceleration vector.

The Boussinesq hypothesis is used to calculate $\nu_t$ from the turbulent kinetic energy $k$ and its specific rate of dissipation $\omega$ according to

$$\nu_t = \frac{k}{\omega}. \tag{3}$$

In REEF3D::CFD, the two-equations $k$-$\omega$ turbulence model [32] is typically applied to propagate the turbulence properties in space and time. Wall functions are taken into account to approximate the boundary layer flow. A limiter for $\nu_t$ is introduced to account for eventual overproduction of turbulence in highly strained flows outside the boundary layer [33]:

$$\nu_t = \min\left(\frac{k}{\omega}, \sqrt{\frac{2}{3}}\frac{k}{|\mathbf{S}|}\right) \tag{4}$$

Special attention is paid to the correct turbulence modelling near the free surface as the turbulent length scales in the water are reduced in its proximity. Standard two-phase RANS turbulence models do not account for this which can lead to increased $\omega$ and damped fluctuations normal to the surface due to a redistributed to parallel fluctuations. Additionally, standard RANS turbulence closure will

incorrectly predict the maximum turbulence intensity at the free surface because the mean rate of strain **S** can be large especially in the vicinity of the interface between water and air [34]. A more realistic representation of the free surface effect on the turbulence can be achieved through the replacement of the original equation for $\omega$ in the vicinity of the surface by the empirical formula [34,35]:

$$\omega_s = \frac{c_\mu^{-0.25}}{\kappa} k^{0.5} \left( \frac{1}{y'} + \frac{1}{y^*} \right),$$ 

(5)

with $c_\mu = 0.07$ and $\kappa = 0.4$. The virtual origin of the turbulent length scale $y'$ is empirically found to be 0.07 times the mean water depth [36]. $y^*$ is the distance from the nearest wall. Hence, a smooth transition from the free surface value to the wall boundary value of $\omega$ is ensured.

The location of the free surface is represented implicitly by the zero level set of a smooth signed distance function $\varphi$ which can be expressed with the Eikonal equation $|\nabla \varphi| = 1$. The simple advection equation

$$\frac{\partial \varphi}{\partial t} + u_j \frac{\partial \varphi}{\partial x_j} = 0$$ 

(6)

is applied for propagating the function in space and time. The hyperbolic property of (6) necessitates the usage of conservative numerical schemes. The level set function has to be reinitialized regularly in order to keep its signed distance property. The PDE-based reinitialization algorithm by Sussman et al. [37] is, therefore, executed after each time step. By solving

$$\frac{\partial \varphi}{\partial \tau} + S(\varphi) \left( \left| \frac{\partial \varphi}{\partial x_j} \right| - 1 \right) = 0,$$ 

(7)

with $\Delta \tau$ an artificial time stepping, the original properties of $\varphi$ can be retained. $S(\varphi)$ is the smoothed sign function [38].

The material properties of the two phases are determined for the whole domain in accordance with the continuum surface force model of Brackbill et al. [39]. The properties are defined at any location in the domain as

$$\rho_i = \rho_w H(\varphi_i) + \rho_a (1 - H(\varphi_i)),$$ 

(8)

$$\nu_i = \nu_w H(\varphi_i) + \nu_a (1 - H(\varphi_i)),$$ 

(9)

with $w$ indicating water and $a$ air properties. $H$ is the smoothed Heaviside step function

$$H(\varphi_i) = \begin{cases} 0 & if \ \varphi_i < -\epsilon \\ \frac{1}{2} \left( 1 + \frac{\varphi}{\epsilon} + \frac{1}{\pi} \sin \left( \frac{\pi \varphi_i}{\epsilon} \right) \right) & if \ |\varphi_i| \leq \epsilon \\ 1 & if \ \varphi_i > \epsilon, \end{cases}$$ 

(10)

Typically, the thickness of the smoothed out interface is chosen to be $\epsilon = 2.1 \Delta x$ on both sides of the interface. The density is generally determined directly at the cell faces in order to avoid spurious oscillations at the interface (see [1] for details).

The numerical discretisation of the different equations is achieved using finite difference methods on rectilinear grids. The coupling of pressure and velocity during the solution of (2) is ensured by staggering the grid. A fifth-order accurate weighted essentially non-oscillatory (WENO) scheme [4] adapted to non-uniform cell sizes is applied for the convection terms. In (6), the convection term is discretised by the fifth-order accurate Hamilton–Jacobi WENO method of Jiang and Peng [40]. Diffusion terms are, generally, discretised using second-order accurate central finite differences.

The solution process follows the projection method for incompressible flows of Chorin [41]. In the predictor step, the conservation equation for momentum (2) is solved without considering the pressure gradients

$$\frac{u_i^{(*)} - u_i^{(n)}}{\Delta t} = -u_j \frac{\partial u_i}{\partial x_j} + \frac{\partial}{\partial x_j} \left( \nu \cdot \left( \frac{\partial u_i}{\partial x_j} + \frac{\partial u_j}{\partial x_i} \right) \right) + g_i. \tag{11}$$

Thus, a predicted velocity field $u_i^{(*)}$ is obtained. Here, the time derivatives are solved by applying the third-order accurate Total Variation Diminishing (TVD) Runge–Kutta scheme [5]. The same time discretisation is also used in (6) and (7). Turbulence time advancement is solved using implicit methods due to its source term driven character. The general time-stepping is controlled adaptively under consideration of the CFL condition (see [1]). Diffusion terms are treated implicitly to overcome their restrictions on this condition. The insertion of the predicted velocities into the continuity equation leads to the Poisson equation

$$\frac{\partial}{\partial x_i} \left( \frac{1}{\rho(\widehat{\Phi}^{n+1})} \frac{\partial p^{(n+1)}}{\partial x_i} \right) = \frac{1}{\Delta t} \frac{\partial u_i^{(*)}}{\partial x_i}. \tag{12}$$

for the pressure of the new time step. It is solved by the fully parallelized BiCGStab algorithm of the HYPRE library [42] with the geometric multigrid PFMG pre-conditioner [43] to enhance the performance. As the final step, the divergence-free velocity field of the new time step is obtained following

$$u_i^{(n+1)} = u_i^{(*)} - \frac{\Delta t}{\rho(\widehat{\Phi}^{n+1})} \frac{\partial p^{(n+1)}}{\partial x_i}. \tag{13}$$

High-performance computations are enabled in REEF3D::CFD by applying the Message Passing Interface (MPI) and ghost cells as the parallelisation strategy. Three layers of ghost cells are added to each sub-domain due to the fifth-order accurate WENO scheme. Similarly, the directional ghost cell immersed boundary method (GCIBM) of Berthelsen and Faltinsen [2] is implemented to handle complex solid geometries. Here, the domain is virtually extended into the geometry, and the values at these ghost cells are found through extrapolation and under consideration of a wall boundary condition. Thus, the numerical discretisation of the fluid domain does not need to account for the boundary conditions explicitly. Instead, they are incorporated implicitly. Simple geometries such as boxes, cylinders or prisms can be generated directly through user input. Otherwise, STL files are to be generated. Then, a level set function, with the zero level set representing the solid boundary, is generated using a ray-tracing algorithm as presented in [44], see above. In the same way, natural bathymetries can be incorporated in a straightforward manner [45].

### 2.2. REEF3D::Sflow

The governing equations for the non-hydrostatic shallow water module are derived from the mass and momentum conservation for an incompressible inviscid fluid. Following the quadratic assumption [46,47], the governing equations are written with depth-averaged variables:

$$\frac{\partial \zeta}{\partial t} + \frac{\partial hu}{\partial x} + \frac{\partial hv}{\partial y} = 0, \tag{14}$$

$$\frac{\partial u}{\partial t} + u\frac{\partial u}{\partial x} + v\frac{\partial u}{\partial y} = -g\frac{\partial \zeta}{\partial x} - \frac{1}{\rho h}\left(\frac{\partial hq}{\partial x} - \left(\frac{3}{2}q + \frac{1}{4}\rho h\Phi_{nh}\right)\frac{\partial d}{\partial x}\right), \tag{15}$$

$$\frac{\partial v}{\partial t} + u\frac{\partial v}{\partial x} + v\frac{\partial v}{\partial y} = -g\frac{\partial \zeta}{\partial y} - \frac{1}{\rho h}\left(\frac{\partial hq}{\partial y} - \left(\frac{3}{2}q + \frac{1}{4}\rho h\Phi_{nh}\right)\frac{\partial d}{\partial y}\right), \tag{16}$$

$$\frac{\partial w}{\partial t} + u\frac{\partial w}{\partial x} + v\frac{\partial w}{\partial y} = -\frac{1}{\rho h}\left(\frac{3}{2}q + \frac{1}{4}\rho h\Phi_{nh}\right), \tag{17}$$

where $u, v, w$ and $q$ are the depth-averaged velocity components in $x, y, z$-directions and the depth-averaged dynamic pressure. $d$ is the still water depth, $\zeta$ represents the free surface elevation and $h = d + \zeta$. The hydrodynamic pressure at the bottom is represented as $\frac{3}{2}q + \frac{1}{4}\rho h\Phi$, which describes the quadratic vertical pressure profile [46]. The term $\Phi$ is expressed as follows [46]:

$$\Phi_{nh} = -\nabla d \cdot (\partial_t \mathbf{u} + (\mathbf{u}\cdot\nabla)\mathbf{u}) - \mathbf{u}\cdot\nabla(\nabla d)\cdot\mathbf{u}. \tag{18}$$

The governing equations are solved on REEF3D's structured staggered grid using finite differences. The solution of the velocities are obtained using Chorin's projection method [41]. The convective terms for the velocities $u$, $v$ and $w$ are discretised with the fifth-order accurate WENO scheme. The TVD third-order accurate Runge–Kutta explicit time scheme is used for time discretisation. The pressure information is obtained from the solution of the Poisson equation

$$\frac{h_p}{\rho}\left(\frac{\partial^2 q}{\partial x^2} + \frac{\partial^2 q}{\partial y^2}\right) + \frac{2q}{\rho h_p} = \frac{1}{\partial x \partial t}\left(-h_p\left(\frac{\partial u}{\partial x} + \frac{\partial v}{\partial y}\right) - 2w - u\frac{\partial d}{\partial x} - v\frac{\partial d}{\partial y}\right). \tag{19}$$

Here, the parameter $h_p$ denotes the water level in the centre of the cell, where the dynamic pressure $q$, the vertical velocities $w$ and the free surface location $\zeta$ are solved. The horizontal velocities $u$ and $v$ are solved at the cell faces. The PFMG preconditioned BiCGStab algorithm [43] of HYPRE is applied to solve for pressure. The solution is then utilised to correct the velocities in a correction step:

$$u^{n+1} = u^* + \Delta t\left(\frac{3}{2}\frac{q^{n+1}}{\rho h_p}\frac{\partial d}{\partial x} + \frac{1}{4}\Phi_{nh}\frac{\partial d}{\partial x}\right), \tag{20}$$

$$v^{n+1} = v^* + \Delta t\left(\frac{3}{2}\frac{q^{n+1}}{\rho h_p}\frac{\partial d}{\partial y} + \frac{1}{4}\Phi_{nh}\frac{\partial d}{\partial y}\right), \tag{21}$$

$$w^{n+1} = w^* + \Delta t\left(\frac{3}{2}\frac{q^{n+1}}{\rho h_p} + \frac{1}{4}\Phi_{nh}\right), \tag{22}$$

with $u^*, v^*, w^*$ the intermediate velocities using only the hydrostatic pressure information.

The free-surface elevation $\zeta$ is determined from Equation (14) using the divergence of the depth-integrated horizontal velocities and the fifth-order WENO scheme.

A straightforward wetting and drying scheme [48,49] is applied at the coastlines. The velocities are set to be zero in cells where the local water level is below a user-defined threshold:

$$\begin{cases} u = 0, & \text{if } \widehat{h}_x < \text{threshold}, \\ v = 0, & \text{if } \widehat{h}_y < \text{threshold}. \end{cases} \tag{23}$$

The default threshold is set to be 0.00005 m. This approach tracks the variations of the coastlines accurately and avoids numerical instabilities by ensuring non-negative water depth [48,49].

Breaking waves are detected when the vertical velocity of the free-surface exceeds a fraction of the shallow water celerity [50]:

$$\frac{\partial \zeta}{\partial t} > \alpha \sqrt{gh}. \tag{24}$$

During breaking, the dynamic pressure is removed at the front of the breaker and only the hydrostatic pressure is present in the momentum equations. Another parameter $\beta$ ($0 < \beta < \alpha$) is introduced to replace $\alpha$ in Equation (24) to stop wave breaking and determine the persistence of the breaking process. $\alpha = 0.6$ and $\beta = 0.3$ are recommended by the SWASH developers [50]. In this combined approach, the momentum is well conserved and the energy is correctly dissipated [50].

*2.3. REEF3D::FNPF*

The governing equation for the fully nonlinear potential flow module REEF3D::FNPF is the Laplace equation [51]

$$\frac{\partial^2 \phi}{\partial x^2} + \frac{\partial^2 \phi}{\partial y^2} + \frac{\partial^2 \phi}{\partial z^2} = 0. \tag{25}$$

Boundary conditions at the free surface and the bottom are required in order to solve for the velocity potential $\phi$. The kinematic and dynamic free surface boundary conditions state that the fluid particles at the free surface must remain at the surface and the pressure at the free surface should be equal to the atmospheric pressure. These boundary conditions can be expressed as follows:

$$\frac{\partial \eta}{\partial t} = -\frac{\partial \eta}{\partial x}\frac{\partial \widetilde{\phi}}{\partial x} - \frac{\partial \eta}{\partial y}\frac{\partial \widetilde{\phi}}{\partial y} + \widetilde{w}\left(1 + \left(\frac{\partial \eta}{\partial x}\right)^2 + \left(\frac{\partial \eta}{\partial y}\right)^2\right), \tag{26}$$

$$\frac{\partial \widetilde{\phi}}{\partial t} = -\frac{1}{2}\left(\left(\frac{\partial \widetilde{\phi}}{\partial x}\right)^2 + \left(\frac{\partial \widetilde{\phi}}{\partial y}\right)^2 - \widetilde{w}^2\left(1 + \left(\frac{\partial \eta}{\partial x}\right)^2 + \left(\frac{\partial \eta}{\partial y}\right)^2\right)\right) - g\eta, \tag{27}$$

where $\eta$ is the free surface elevation, $\mathbf{x} = (x, y)$ represents the horizontal directions, $\widetilde{\phi} = \phi(\mathbf{x}, \eta, t)$ and $\widetilde{w}$ are the velocity potential and the vertical velocity at the free surface. At the bottom, the component of the velocity normal to the bottom surface must be zero at all times. This gives the bottom boundary condition

$$\frac{\partial \phi}{\partial z} + \frac{\partial h}{\partial x}\frac{\partial \phi}{\partial x} + \frac{\partial h}{\partial y}\frac{\partial \phi}{\partial y} = 0, \quad z = -h, \tag{28}$$

with $h = h(\mathbf{x})$ the water depth measured from the still water level to the bottom.

The Laplace equation is solved in each time step with the finite difference method on a $\sigma$-coordinate system as proposed by Li and Fleming [52]. Here, the $\sigma$-coordinate system follows the irregular variation of the water depth. A Cartesian grid can be transformed to a $\sigma$-coordinate as follows:

$$\sigma = \frac{z + h(\mathbf{x})}{\eta(\mathbf{x}, t) + h(\mathbf{x})}. \tag{29}$$

The vertical coordinates are clustered towards the free surface by including a stretching function:

$$\sigma_i = \frac{\sinh(-\alpha) - \sinh\left(\alpha\left(\frac{i}{N_z} - 1\right)\right)}{\sinh(-\alpha)}, \tag{30}$$

where $\alpha$ is the stretching factor, $i$ is the index of the vertical grid point and $N_z$ stands for the total number of cells in the vertical direction. The boundary conditions and the governing equation in the $\sigma$-coordinate can be written as:

$$\Phi = \widetilde{\phi} \qquad\qquad , \sigma = 1; \tag{31}$$

$$\frac{\partial^2 \Phi}{\partial x^2} + \frac{\partial^2 \Phi}{\partial y^2} + \left( \frac{\partial^2 \sigma}{\partial x^2} + \frac{\partial^2 \sigma}{\partial y^2} \right) \frac{\partial \Phi}{\partial \sigma} + 2 \left( \frac{\partial \sigma}{\partial x} \frac{\partial}{\partial x} \left( \frac{\partial \Phi}{\partial \sigma} \right) + \right.$$
$$\left. \frac{\partial \sigma}{\partial y} \frac{\partial}{\partial y} \left( \frac{\partial \Phi}{\partial \sigma} \right) \right) + \left( \left( \frac{\partial \sigma}{\partial x} \right)^2 + \left( \frac{\partial \sigma}{\partial y} \right)^2 + \left( \frac{\partial \sigma}{\partial z} \right)^2 \right) \frac{\partial^2 \Phi}{\partial \sigma^2} = 0 \quad , 0 \leq \sigma < 1; \tag{32}$$

$$\left( \frac{\partial \sigma}{\partial z} + \frac{\partial h}{\partial x} \frac{\partial \sigma}{\partial x} + \frac{\partial h}{\partial y} \frac{\partial \sigma}{\partial y} \right) \frac{\partial \Phi}{\partial \sigma} + \frac{\partial h}{\partial x} \frac{\partial \Phi}{\partial x} + \frac{\partial h}{\partial y} \frac{\partial \Phi}{\partial y} = 0 \qquad , \sigma = 0, \tag{33}$$

with $\Phi$ the velocity potential with a dependency on $\sigma$. The fluid velocities can then be calculated using

$$u\left(\mathbf{x}, z\right) = \frac{\partial \Phi\left(\mathbf{x}, z\right)}{\partial x} = \frac{\partial \Phi\left(\mathbf{x}, \sigma\right)}{\partial x} + \frac{\partial \sigma}{\partial x} \frac{\partial \Phi\left(\mathbf{x}, \sigma\right)}{\partial \sigma}, \tag{34}$$

$$v\left(\mathbf{x}, z\right) = \frac{\partial \Phi\left(\mathbf{x}, z\right)}{\partial y} = \frac{\partial \Phi\left(\mathbf{x}, \sigma\right)}{\partial y} + \frac{\partial \sigma}{\partial y} \frac{\partial \Phi\left(\mathbf{x}, \sigma\right)}{\partial \sigma}, \tag{35}$$

$$w\left(\mathbf{x}, z\right) = \frac{\partial \Phi\left(\mathbf{x}, z\right)}{\partial z} = \frac{\partial \sigma}{\partial z} \frac{\partial \Phi\left(\mathbf{x}, \sigma\right)}{\partial \sigma}. \tag{36}$$

The Laplace equation is discretized using second-order central differences, and the solution is obtained using the geometric multigrid preconditioned conjugated gradient solver provided by HYPRE. The convection terms in the free surface boundary conditions are discretized using the fifth-order accurate Hamilton–Jacobi version of the WENO scheme [40]. The time-dependent terms in the free surface boundary conditions are treated with the third-order accurate TVD Runge–Kutta scheme [5]. An adaptive time step is included by controlling a constant time factor that is equivalent to the Courant criterion [53]:

$$c_u = \frac{\Delta x}{\left| \max(u_{\max}, \sqrt{9.81 * d_{\max}}) \right|},$$
$$c_v = \frac{\Delta x}{\left| \max(v_{\max}, \sqrt{9.81 * d_{\max}}) \right|}, \tag{37}$$
$$c_{\text{tot}} = \min(c_u, c_v),$$
$$\Delta t = c_{\text{tot}} \text{ CFL,}$$

where $c_u, c_v, c_w$ are the phase velocities in $x$, $y$ and $z$ directions, and $u_{\max}, v_{\max}$ are the maximum particle velocities in $x$- and $y$-direction.

The wetting-drying scheme for detecting coastlines and the shallow water breaking criterium follow the same principle as in REEF3D::SFLOW. For deep water breaking, a wave slope criterion is used. Wave breaking takes place when the ratio between the free surface elevation difference and the horizontal distance difference at adjacent cells is higher than the criterion, which has a default value of 1.25. A filtering scheme is used to smooth the free surface in order to dissipate wave energy when wave breaking is detected [54].

Another challenge in handling coastlines in a potential flow model is the possible numerical instability during the wave run-up process in the swash zone. The derivatives of velocity potential over water depth in Equation (32) indicate a possible numerical instability when water depth becomes infinitesimal. Therefore, an innovative coatline lagorithm is introduced to eliminate the instability.

After the wet and dry cells are identified, the wet cells are assigned with $+1$ and the dry cells are assigned with $-1$. With these initial values, the coastline is captured using the level-set function by Osher and Sethian [3]:

$$\varphi(\vec{x}, t) \begin{cases} > 0 \; if \; \vec{x} \in wet \; cell \\ = 0 \; if \; \vec{x} \in \Gamma \\ < 0 \; if \; \vec{x} \in dry \; cell \end{cases} \tag{38}$$

$\Gamma$ represents the coastline, and the Eikonal equation $|\nabla \varphi| = 1$ remains valid in the level-set function. From the initial values, the correct signed distance function is obtained by solving the following Partial Differential Equation (PDE) based reinitialisation function [37]:

$$\frac{\partial \varphi}{\partial t} + S(\varphi) \left( \left| \frac{\partial \varphi}{\partial x_j} \right| - 1 \right) = 0 \tag{39}$$

where $S(\varphi)$ is the smoothed sign function [38]. This equation is solved until convergence and results in the correct signed distance away from the coastline in the whole horizontal plane. The exact coastline location is the zero-contour of the level set function.

Relaxation zones are applied along the the wet side of the coastline. With these relaxation zones, the extreme run-ups are avoided and therefore eliminate numerical instabilities in the free surface boundary conditions at extreme shallow regions.

## 3. Numerical Results

### 3.1. Comparison of the Different Modules for the Numerical Simulation of Progressive Waves

The different modules of REEF3D all share high-order numerical schemes for spatial and temporal discretisation and a high-performance computation capacity. To demonstrate the modules' capabilities and limitations, simulations of progressive waves over constant and varying topography are performed using all three modules. First, progressive regular wave propagation over constant intermediate water depth in 2D is simulated. The numerical wave tank is 28 m long and the water depth is 0.5 m. Two input waves are used, one is a linear wave with the wave height $H = 0.01$ m and a wave period of $T = 1.95$ s, and another is a Stokes 2nd-order wave with a wave height of $H = 0.05$ m and the same wave period of $T = 1.95$ s and wavelength 3.936 m. A one-wavelength wave generation zone is located at the inlet boundary, and a two-wavelength numerical beach is arranged at the outlet boundary. All simulations are conducted for a duration of 40 s on a Mac Pro with a four 2.7 GHz Intel Xeon E5 cores. The grid convergence studies of the linear wave simulations are shown in Figure 1a–c. For REEF3D::FNPF, the vertical grid is determined by keeping a constant truncation error in the vertical direction [55], which results in 10 vertical cells with a stretching factor of 1.25. It is seen that the results for amplitude and phase converge with $\Delta x = 0.05$ m, 0.02 m and 0.1 m for REEF3D::CFD, REEF3D::SFLOW and REEF3D::FNPF, respectively. With these cell sizes, the total number of cells $N_t$ and the simulation time $T_s$ are compared in Table 1. The spatial free surface profiles are compared against the theoretical wave profile in Figure 1d. All three modules generate the theoretical wave profile accurately and the numerical beach absorbs the wave energy at the outlet boundary effectively. REEF3D::SFLOW requires the least number of cells due to its two-dimensional grid. Consequently, it is 7.3 times faster as REEF3D::CFD. However, REEF3D::FNPF is the fastest (35 times as fast at REEF3D::CFD), even though it needs more cells than REEF3D::SFLOW.

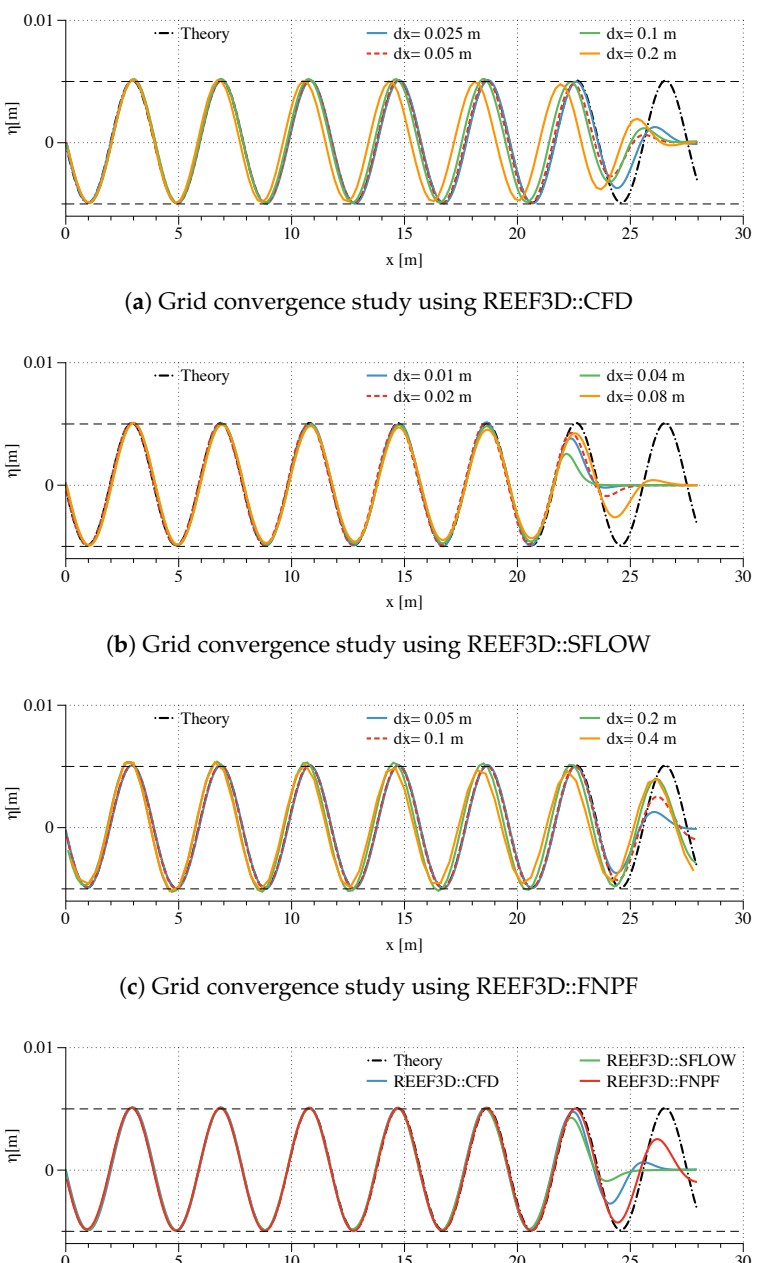

(**a**) Grid convergence study using REEF3D::CFD

(**b**) Grid convergence study using REEF3D::SFLOW

(**c**) Grid convergence study using REEF3D::FNPF

(**d**) Comparison of the spatial wave profiles

**Figure 1.** Convergence study on cell sizes for the 2D regular linear wave simulation and the comparison of free surface elevation among the three modules. (**a**–**c**) grid convergence study, (**d**) comparison of the spatial wave profiles using the finest cell sizes.

The mean square root errors for wave height in the grid convergence study for the 2D regular linear wave simulation using the three modules are summarised in Table 2.

**Table 1.** Comparison of total number of cells $N_t$ and simulation time $T_s$ in seconds for the simulation of progressive linear wave using the three modules.

| Module | $N_t$ | $T_s$ |
|---|---|---|
| REEF3D::CFD | 11,200 | 594.9 s |
| REEF3D::SFLOW | 560 | 81.5 s |
| REEF3D::FNPF | 2800 | 16.8 s |

**Table 2.** Mean square root errors on wave height in the grid convergence study for the 2D regular linear wave simulation using the three modules. The notations dx1 to dx4 represent the finest and coarsest cell size in the tests of each of the modules.

| dx (m) | REEF3D::CFD | REEF3D::SFLOW | REEF3D::FNPF |
|--------|-------------|---------------|--------------|
| dx1 | $7.889 \times 10^{-5}$ | $8.031 \times 10^{-5}$ | $5.025 \times 10^{-5}$ |
| dx2 | $8.872 \times 10^{-5}$ | $9.656 \times 10^{-5}$ | $5.701 \times 10^{-5}$ |
| dx3 | $1.010 \times 10^{-4}$ | $1.999 \times 10^{-4}$ | $3.303 \times 10^{-4}$ |
| dx4 | $1.213 \times 10^{-4}$ | $4.251 \times 10^{-4}$ | $4.842 \times 10^{-4}$ |

Similarly, the grid convergence study and the comparison of the spatial wave profiles for the simulations of the 2nd-order Stokes wave using different modules are shown in Figure 2. The mean square root errors for wave height in the grid convergence study for the 2D regular Stokes 2nd-order wave simulation using the three modules are summarised in Table 3. It is seen that the grid convergence is achieved with $\Delta x = 0.05$ m, 0.02 m and 0.1 m for REEF3D::CFD, REEF3D::SFLOW and REEF3D::FNPF. With these cell sizes, all three modules represent the 2nd-order Stokes wave with correct amplitude, phase and asymmetry over the still water level. Similarly, the total number of cells and computational time are summarised in Table 4, and the computational speed is similar to the linear wave simulations.

**Table 3.** Mean square root errors for wave height in the grid convergence study for the 2D regular Stokes 2nd-order wave simulation using the three modules. The notations dx1 to dx4 represent the finest and coarsest cell size in the tests of each of the modules.

| dx (m) | REEF3D::CFD | REEF3D::SFLOW | REEF3D::FNPF |
|--------|-------------|---------------|--------------|
| dx1 | $3.581 \times 10^{-4}$ | $5.117 \times 10^{-4}$ | $4.739 \times 10^{-4}$ |
| dx2 | $3.582 \times 10^{-4}$ | $7.637 \times 10^{-4}$ | $5.483 \times 10^{-4}$ |
| dx3 | $4.421 \times 10^{-4}$ | $9.529 \times 10^{-4}$ | $1.41 \times 10^{-3}$ |
| dx4 | $1.109 \times 10^{-3}$ | $1.80 \times 10^{-3}$ | $2.15 \times 10^{-3}$ |

**Table 4.** Comparison of total number of cells $N_t$ and simulation time $T_s$ in seconds for the simulation of progressive 2nd-order Stokes wave using the three modules.

| Module | $N_t$ | $T_s$ |
|--------|-------|-------|
| REEF3D::CFD | 11,200 | 638.3 s |
| REEF3D::SFLOW | 560 | 86.7 s |
| REEF3D::FNPF | 2800 | 16.9 s |

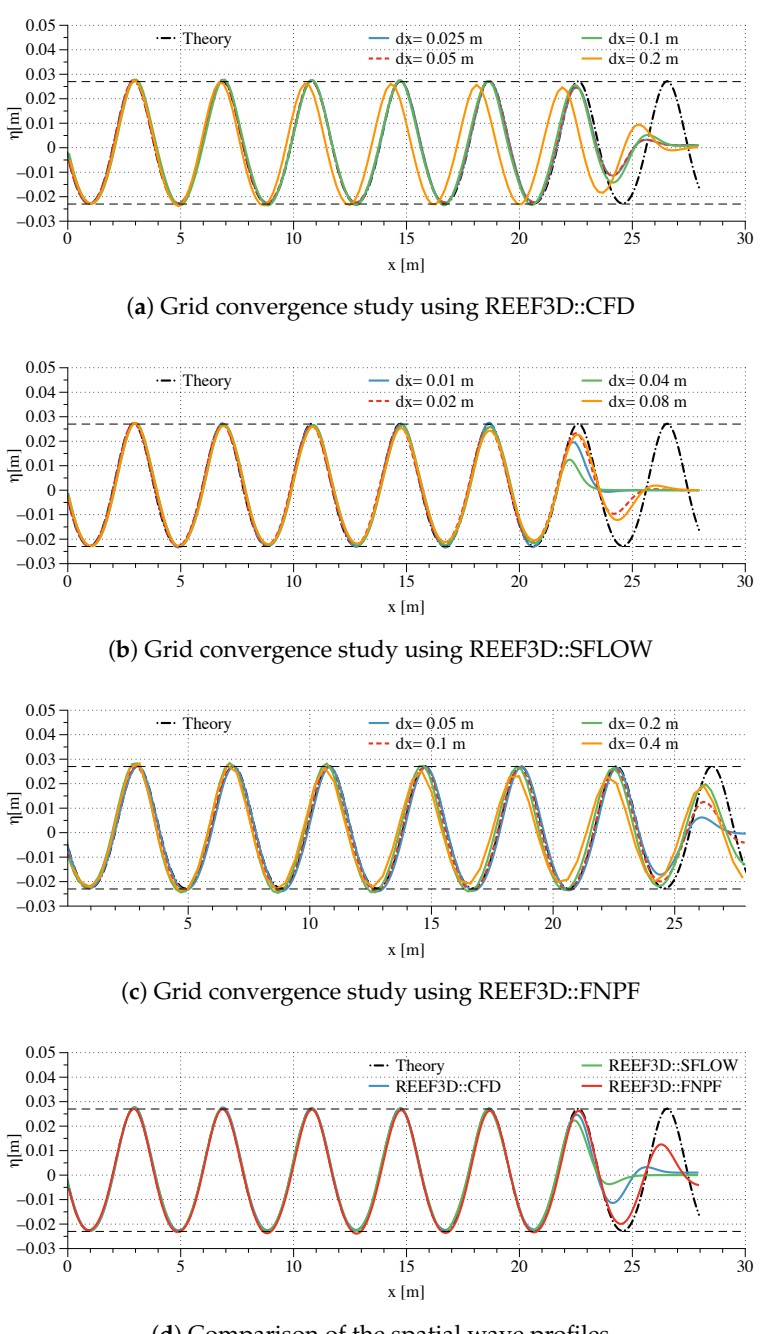

(**a**) Grid convergence study using REEF3D::CFD

(**b**) Grid convergence study using REEF3D::SFLOW

(**c**) Grid convergence study using REEF3D::FNPF

(**d**) Comparison of the spatial wave profiles

**Figure 2.** Convergence study on cell sizes for the 2D regular Stokes 2nd-order wave simulation and the comparison of free surface elevation among the three modules. (**a**–**c**) grid convergence study, (**d**) comparison of the spatial wave profiles using the cell sizes achieving grid convergence.

### 3.2. Two-Dimensional Wave Propagation over a Submerged Bar

Next, the experiment of the wave propagation over a submerged bar [56] is reproduced using all three modules. The numerical tank setup is shown in Figure 3. A wave generation zone of 5 m is located at the inlet boundary and a numerical beach of 9.5 m is located at the outlet boundary. The submerged bar starts 6 m from the wave generation zone, and eight wave gauges are located over the horizontal range of the submerged bar. A 2nd-order Stokes wave with a wave height 0.021 m and a wave period of 2.525 s is generated from the inlet boundary and propagates over the bar for 60 s. The simulations are computed with four 2.7 GHz Intel Xeon E5 cores on Mac Pro for REEF3D::FNPF and REEF3D::SFLOW and 128 2.1 GHz Intel E5-2683v4 cores on the supercomputer Fram for REEF3D::CFD.

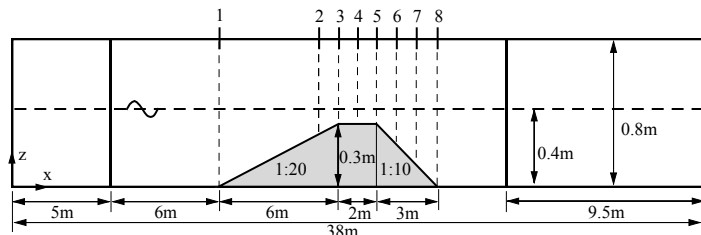

**Figure 3.** Numerical setup for the simulation of the wave propagation over a submerged bar.

The grid convergence study is shown in Figure 4. The vertical grid arrangement for REEF3D::FNPF follows the same constant truncation error principle. Here, 10 vertical cells and a stretching factor of 1.2 is used. Only the horizontal grid convergence of REEF3D::FNPF is performed. The last wave gauge 8 is used for the convergence study as high-frequency wave components appear during the de-shoaling process after the waves propagate over the submerged bar. REEF3D::CFD and REEF3D::FNPF are able to capture the high-frequency wave components with cell sizes of 0.005 m and 0.025 m, respectively. For REEF3D::SFLOW, even with a converged cell size of 0.02 m, the wave phases are not correctly represented because these high-frequency waves have significantly shorter wavelengths and the water condition is not appropriate for shallow water models at this location.

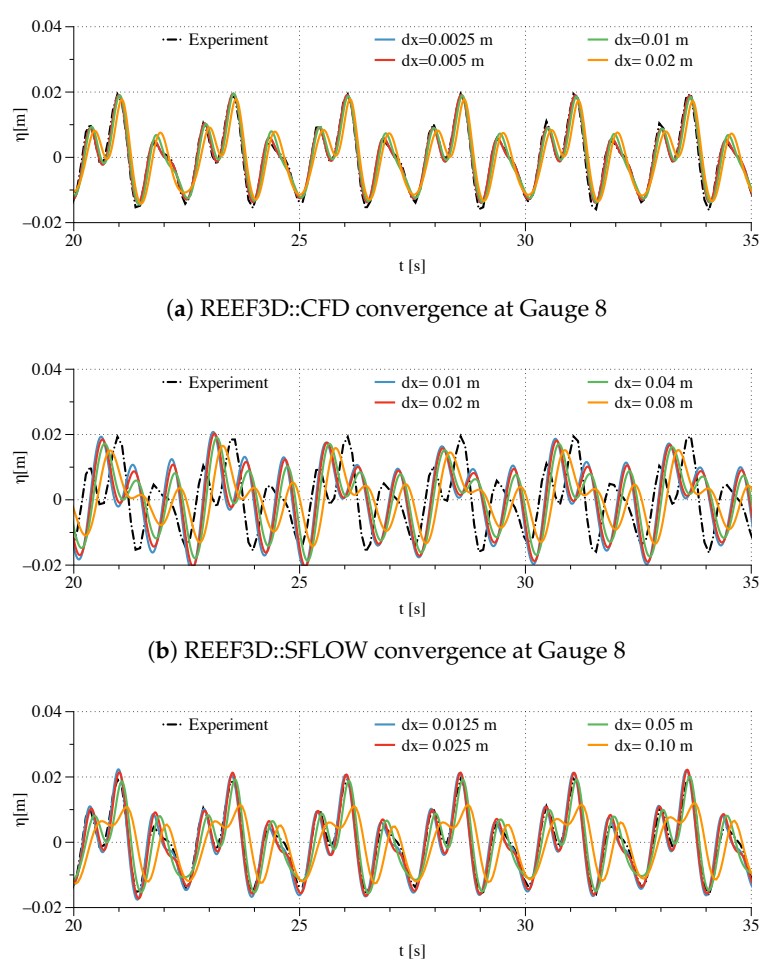

**Figure 4.** Convergence study on horizontal cell sizes at wave gauge 8 for the simulations of wave propagation over a submerged bar. (**a**) REEF3D::CFD grid convergence, (**b**) REEF3D::SFLOW grid convergence, (**c**) REEF3D::FNPF grid convergence.

Using the converged cell sizes, the free surface elevation time history in the simulations are compared against the experimental measurements in Figure 5. The free surfaces from all simulations agree well with the experimental data during the shoaling process, while REEF3D::SFLOW starts to show phase differences from gauge 6 in the de-shoaling process as the water condition gets deeper due to shorter waves.

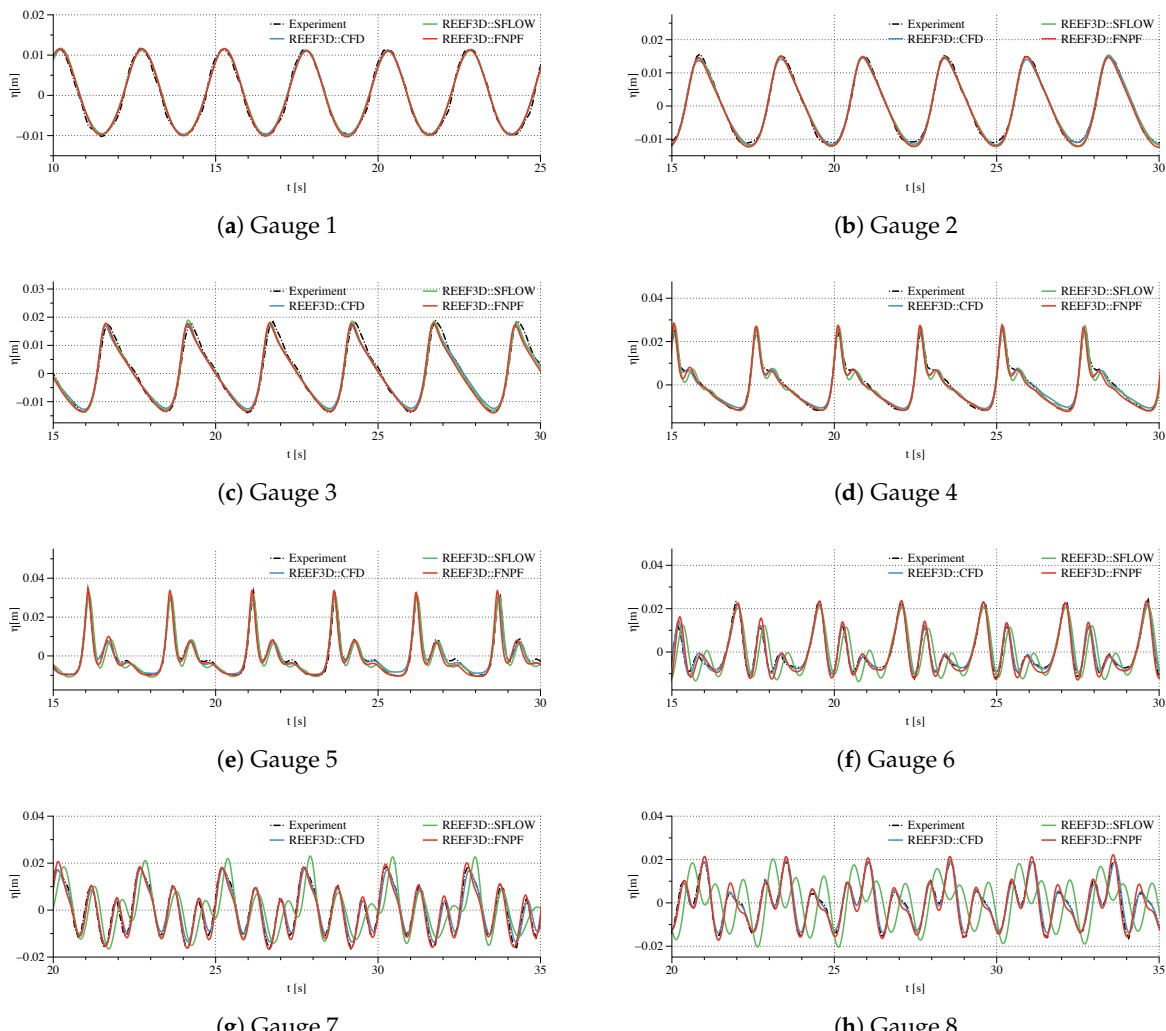

**Figure 5.** Comparison of the time histories of the free surface elevations at the wave gauges in the simulations of wave propagation over a submerged bar using the cell sizes achieving grid convergence.

The number of cells and computational time for the simulations of the wave propagation over a submerged bar are summarised in Table 5. When complicated phenomena are present, CFD often requires a large number of cells, and the speed-up with the shallow water model and the potential flow model is dramatically increased.

**Table 5.** Comparison of total number of cells $N_t$ and simulation time $T_s$ in seconds for the simulation of wave propagation over a submerged bar using the three modules.

| Module | $N_t$ | $T_s$ |
|---|---|---|
| REEF3D::CFD | 1,216,000 | 10,759.5 s |
| REEF3D::SFLOW | 1900 | 761.7 s |
| REEF3D::FNPF | 15,200 | 282.2 s |

The simulations show that, for progressive regular waves below the breaking limit, all three modules can represent the frees surface accurately. However, the requirements of the grid resolution are different. It is commonly seen that 80 to 100 cells per wavelength is able to capture the free-surface well with REEF3D::CFD, while only 30 to 40 cells per wavelength are needed in REEF3D::FNPF. The grid resolution in REEF3D::SFLOW might be higher, but the 2D vertical grid structure reduces the total number of cells dramatically. In practice, when the wave steepness is not close to the breaking limit, REEF3D::SFLOW and REEF3D::FNPF are much faster alternatives, especially for large-scale sea states and coastal wave simulations. In shallow for intermediate water condition up to wavelength to water depth ratio 0.25 [46], REEF3D::SFLOW has an advantage because it is capable of resolving the run-up process in the swash zone. However, for water conditions with large water depth changes, the de-shoaling process limits the application of REEF3D::SFLOW as seen in the simulation of wave propagation over a submerged bar. In such conditions, REEF3D::FNPF is the optimal choice as its applicability is not limited by large water depth gradients. REEF3D::CFD is slower but contains more information about turbulent effects in the flow. In cases where strong wave–structure interaction take place or waves break, REEF3D::CFD is the only option for numerical modelling of the associated phenomena. The following applications focus on the most suitable applications for each of the modules.

### 3.3. Two-Dimensional Wave Breaking over a Mild Slope

In shallow water regions, depth-induced wave breaking is a common phenomenon. All three modules are equipped with breaking wave algorithms to represent the energy dissipation during a wave breaking process, as described in Section 2. In this section, a depth-induced breaking wave over a mild slope is simulated with all three modules in a two-dimensional numerical wave tank. In order to reduce the computational cost of the CFD simulation, the original setup from Ting and Kirby [57] is truncated in its longitudinal direction. The breaking wave zone and swash zone are all remained in the truncated numerical wave tank. The new numerical wave tank setup is shown in Figure 6. The mild slope starts 13.8 m from the inlet boundary and rises up to 0.463 m at the outlet following a slope of 1:35. The water depth at the wave generator is 0.4 m. A 5*th*-order Cnoidal wave with a wave height of 0.128 m and wave period of 5 s is generated at wave generation zone that is 9.8 m long, i.e., one wavelength. Four wave gauges are located on the slope adjacent to the wave breaking location. From wave gauges 1 to 4, the x-coordinates are $x = 19.8, 20.8, 21.8$ and 22.1 m. The simulations are computed with four 2.7 GHz Intel Xeon E5 cores on Mac Pro for REEF3D::FNPF and REEF3D::SFLOW and 128 2.1 GHz Intel E5-2683v4 cores on the supercomputer Fram. The grid convergence study for the three models REEF3D::CFD, REEF3D::SFLOW and REEF3D::FNPF were reported respectively by Bihs et al. [1], Wang et al. [47] and Bihs et al. [51]. As a result, the $dx = 0.005$ m, $dx = 0.005$ m and $dx = 0.005$ m are used in the REEF3D::CFD, REEF3D::SFLOW and REEF3D::FNPF simulations respectively. Ten cells are used in the vertical direction for the simulation with REEF3D::FNPF. The simulations are performed for 40 s with adaptive time stepping and $CFL = 0.1, 0.2$ and 1.0 for the REEF3D::CFD, REEFD::SFLOW and REEF3D::FNPF simulations, respectively. The simulated free surface elevation time series from all three modules are compared to the experimental measurements in Figure 7.

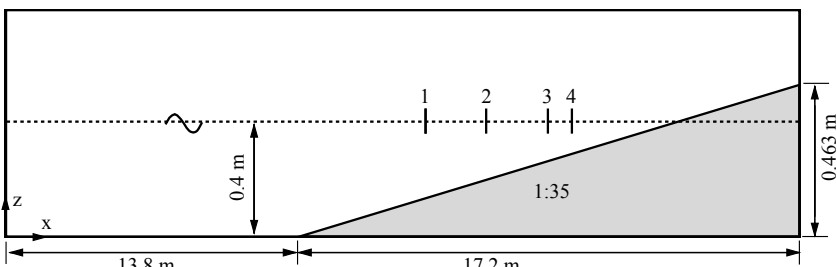

**Figure 6.** Numerical wave tank setup for wave breaking over a mild slope.

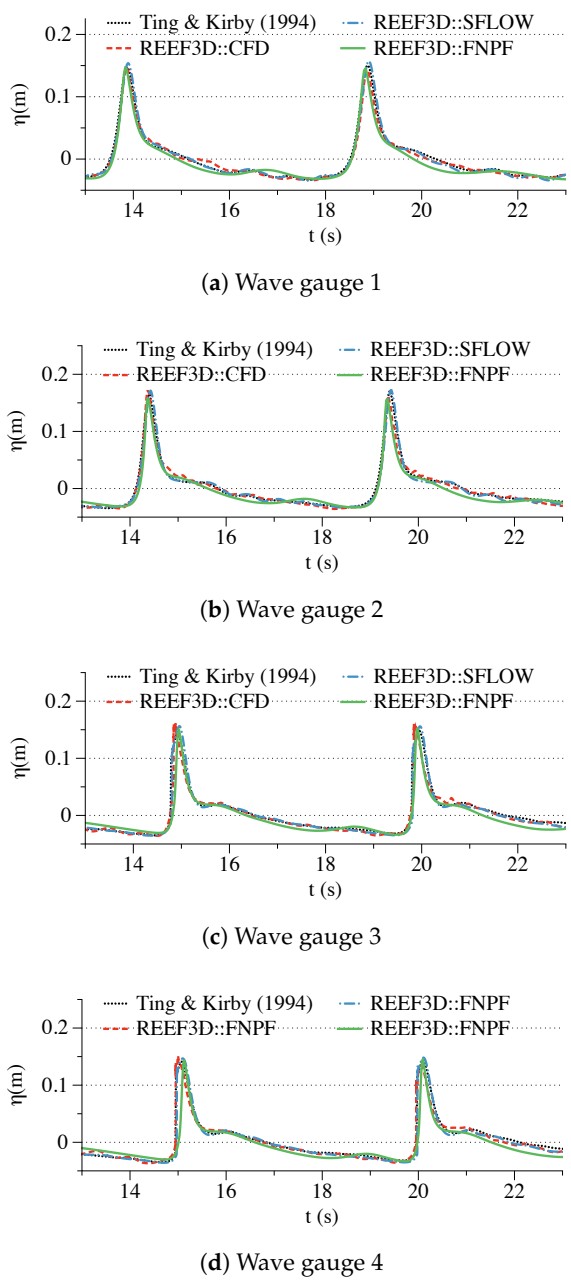

(**a**) Wave gauge 1

(**b**) Wave gauge 2

(**c**) Wave gauge 3

(**d**) Wave gauge 4

**Figure 7.** Comparison between the simulated free surface elevation time series from the three REEF3D modules and the experiment measurements at all four wave gauges in the simulations of wave breaking over a mild slope.

As can be seen in Figure 7, the results from all three modules achieve a good agreement with the experiment both in wave amplitude and wave phase. The wave amplitudes increase from waver gauge 1 to wave gauge 2 due to the shoaling effect in both the simulations and the experiment. Wave gauge 3 shows a decrease in wave amplitude and the decreasing trend continues to wave gauge 4. This change of amplitude indicates a wave breaking happens between wave gauge 2 and 3 as a result of energy dissipation during the wave breaking process. The correct representation of the amplitude change shows that all three modules produce correct wave energy dissipation.

To compare the computational performance of the three modules, the total number of cells and computational time for each model to finish the simulations are summarised in Table 6.

**Table 6.** Comparison of total number of cells $N_t$ and simulation time $T_s$ in seconds for the simulation of wave propagation over a submerged bar using the three modules.

| Module | $N_t$ | $T_s$ |
|---|---|---|
| REEF3D::CFD | 1,200,000 | 31,578.8 s |
| REEF3D::SFLOW | 6000 | 5326.62 s |
| REEF3D::FNPF | 6000 | 639.9 s |

Similar to Section 3.2, REEF3D::SFLOW and REEF3D::FNPF use much less cells in comparison to REEF3D::CFD to achieve a similar level of accuracy. In this case, both REEF3D::SFLOW and REEF3D::FNPF only need 1/200 the number of cells as used in the REEF3D::CFD simulation. In terms of the computational speed, REEF3D::SFLOWS is seen to be roughly 190 times faster than REEF3D::CFD while REEF3D::FNPF is 1580 times faster. However, the slower computational speed of REEF3D::CFD is compensated by the fact that REEF3D::CFD is the only model that is able to represent a correct geometry of an overturning breaker, which is shown in the next section with a three-dimensional overturning wave breker.

### 3.4. Three-Dimensional Wave Breaking over a Flat-Tipped Reef

The design of coastal structures such as combined coastal defences, recreational surfing reefs and marine biodiversity enhancement measures such as submerged porous reefs require a detailed analysis of the interaction between the incident waves and the proposed structure. The evaluation of the properties of the breaking waves generated due to the presence of the submerged structure is one of the essential analyses in such cases. In this sub-section, three-dimensional wave breaking is investigated using all three models. The free surface elevations at different locations calculated by the two models are also compared. The illustration of the numerical wave tank with the bottom topography used in the simulations is presented in Figure 8. The bottom topography consists of a 1 in 20 slope over which a flat-tip shaped reef with a reef slope of 1 in 6 is placed. The reef angle that is the angle between the reef normal and the direction of wave propagation is 60°. A detailed description of the complicated reef geometry is provided in [58]. The width of the flat tip is 0.188 m and the width of the reef at the far end is 3.88 m. The numerical wave tank is 20 m long, 9 m wide, 0.8 m wide with a water depth of $d = 0.4$ m. Cnoidal waves with a height of $H = 0.12$ m and period $T = 2.50$ s are generated. The submerged reef will affect the propagation of the incident waves and induce wave breaking with the overturning wave crest first appearing over the slope of the reef as shown in Figure 9. The rest of the wavefront undergoes overturning as it propagates further along the submerged reef and the bottom slope. All simulations are computed with 128 2.1 GHz Intel E5-2683v4 cores on the supercomputer Fram.

The free surface elevations at different locations along the reef in the numerical wave tank using the three models are presented in Figure 10. The incident wave at the toe of the slope near the wall is shown in Figure 10a. The free surface elevation over the reef slope is seen in Figure 10b,c. The wave appears to break at these locations as seen from the vertical wave crest front. The difference between the results from the two models are seen in the shape of the wave crest front. The shallow water model, REEF3D::SFLOW and the potential flow model REEF3D::FNPF cannot account for an overturning crest and therefore represent a perfectly vertical wave crest fronts to represent the breaking wave before a sudden reduction in the free surface elevation. In the time series in Figure 10b,c, this is seen through the graph retracing its path, before its eventual reduction. In contrast, REEF3D::CFD represents the overturning wave crest. Therefore, the vertical wave crest front is followed by a reduction of the free surface elevation, without a period of retracing of the initial path to the peak. The wave gauges WG 2, 3 and 4 show this process in Figure 10b–d respectively as they are placed in the region of wave breaking over the reef slope. The free surface elevations at WG 5, 6 and 7 in Figure 10e–g respectively show the secondary breaking process and the post breaking splash up. This is signified by the reduced free surface elevations and the appearance of secondary crests in the time series. A slight phase

difference is seen between the results from REEF3D::SFLOW and REEF3D::CFD. The first secondary breaker in the REEF3D::FNPF simulation is in phase with the other two models. However, significant phase differences are seen in comparison to the other two models after the first secondary breaking. The reason is that the incoming waves start to interact with the wave run-up and run-down on the slope which takes place after the first secondary breaker. In the potential flow model, the wet side of the coastline is covered with a narrow relaxation zone of 0.675 m to avoid numerical instabilities due to the derivatives of the velocity potential over z in the infinitesimal water depth. Therefore, the run-up and run-down are not correctly represented, which lead to a large phase different and smaller wave amplitude in the potential flow simulation. The complex 3D swash zone dynamic and the steeper slope at the end of the numerical wave tank amplify this effect, which is not noticeable in Section 3.3. Figure 10h–j present the free surface elevations at WG 8, 9 and 10, respectively, which are along the reef slope but in post-breaking region. The free surface elevations are seen to be further reduced and several secondary crests appear in the time series. There is also some phase difference seen among the models. On the other hand, the wave heights calculated by all models are similar for the first breaking wave. This suggests that the loss of wave energy due to wave breaking is well represented in the shallow water model as well as the potential flow model, even though the overturning wave crest is not accounted for.

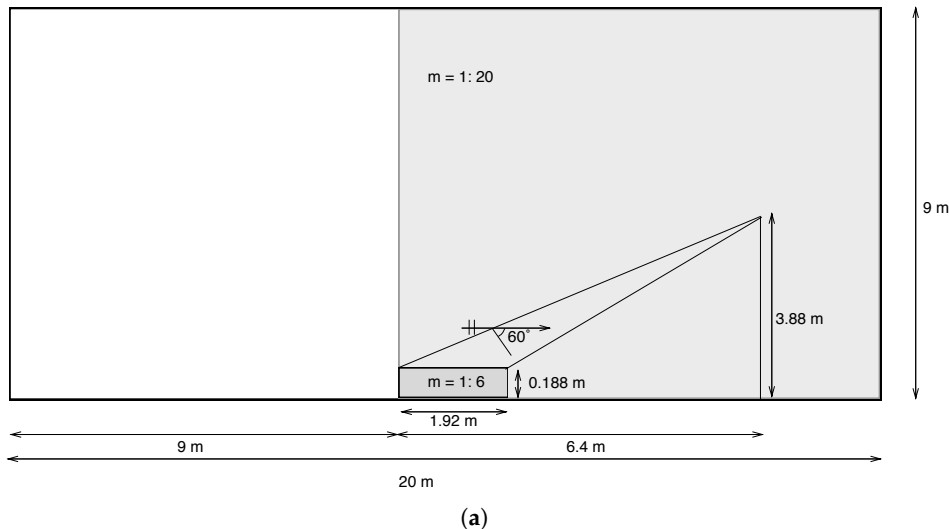

**(a)**

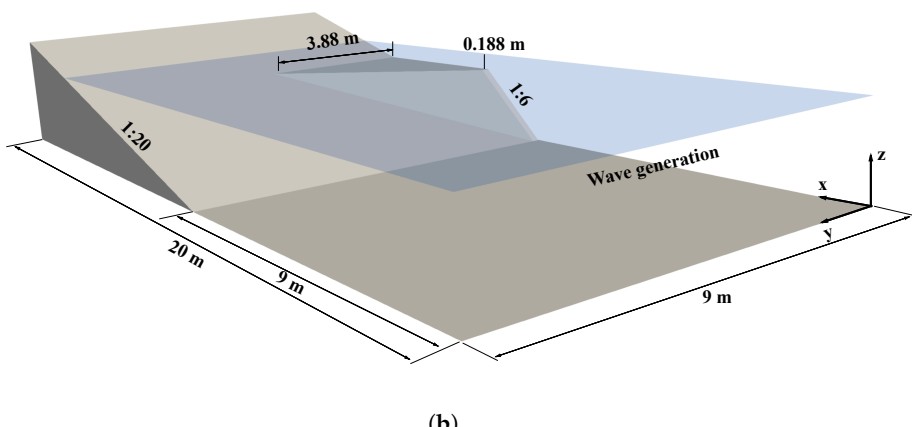

**(b)**

**Figure 8.** Numerical wave tank setup for the simulation of three-dimensional wave breaking on a reef. *m* represents the slopes. (**a**) schematics from top view, (**b**) 3D view in the NWT.

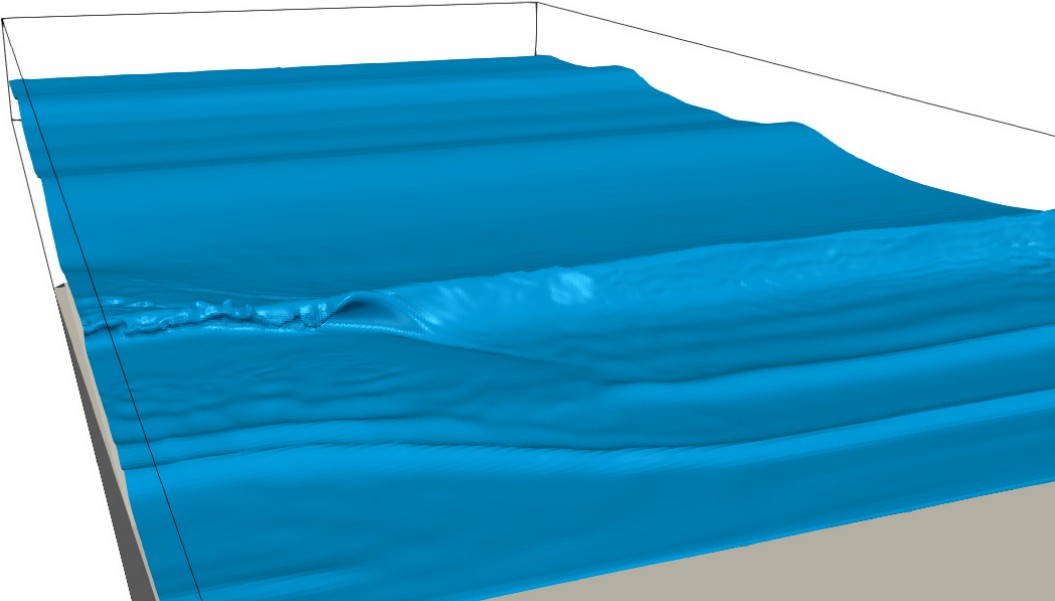

**Figure 9.** Three-dimensional wave breaking over the reef in the numerical wave tank calculated using REEF3D::CFD.

The free surface elevations in the numerical wave tank with the horizontal velocity contours for the simulations carried out using all three models are presented in Figure 11. The overturning wave crest at $t/T = 5.5$ is represented in the CFD model in Figure 11a, whereas only a steep free surface is seen in REEF3D::SFLOW and REEF3D::FNPF in Figure 11c,e. The free surface and velocities over the rest of the wavefront are seen to be similar for all the models. The overturning wave crest moves towards the preceding wave trough and the rest of the wavefront gets steeper at $t/T = 5.6$ in Figure 11b in the REEF3D::CFD model. The REEF3D::SFLOW and REEF3D::FNPF simulations show smoothened free surfaces in the region of the overturning wave crest in Figure 11d,f. Wave breaking is seen on the reef slope and wave breaking is initiated away from the reef in Figure 11g at $t/T = 5.8$ in the REEF3D::CFD simulation. Figure 11i,k show steep wavefronts in the region away from the reef for the REEF3D::SFLOW and REEF3D::FNPF simulations. The process of secondary breaking is seen to have started at this time step in the simulations. The overturning wave crest in the region away from the reef at $t/T = 6.1$ is seen in Figure 11h in the REEF3D::CFD simulation. The free surfaces in the REEF3D::SFLOW ad REEF3D::FNPF simulations in Figure 11j.l are seen to be similar over the reef in the absence of wave breaking and a steep wavefront are seen away from the reef. However, the post-breaking region is seen to be very different in the simulation of REEF3D::FNPF in comparison to the other models, as seen in Figure 11k,l. Less run-up on the slope and some small high-frequency waves are seen only in the simulation of REEF3D::FNPF as the result of the coastal relaxation zone arrangement.

The key difference in the results from REEF3D::CFD and the other two models is that the overturning wave crest is not represented by REEF3D::SFLOW and REEF3D::FNPF. On the other hand, the wave heights after the wave breaking process are seen to be similar in all models. Therefore, if the representation of the overturning wave crest is not critical in a simulation, the shallow water model and potential flow model can provide similar wave kinematics solutions as the three-dimensional and two-phase flow model. However, REEF3D::SFLOW is a better choice when swash zone dynamics result in strong interaction with the incoming waves.

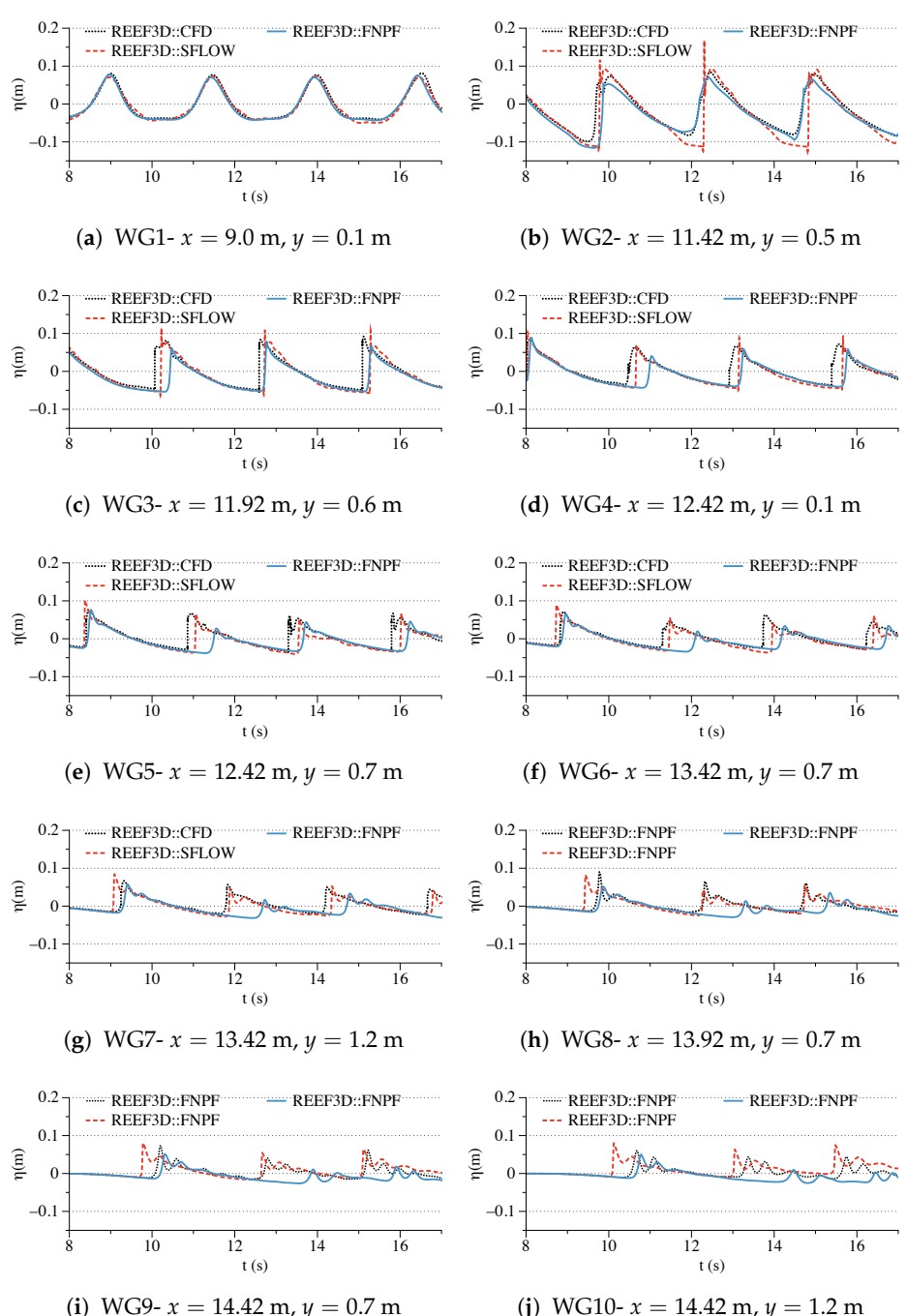

**Figure 10.** Free surface elevations at several locations in the numerical wave tank for three-dimensional wave breaking on a submerged reef calculated using CFD and SFLOW.

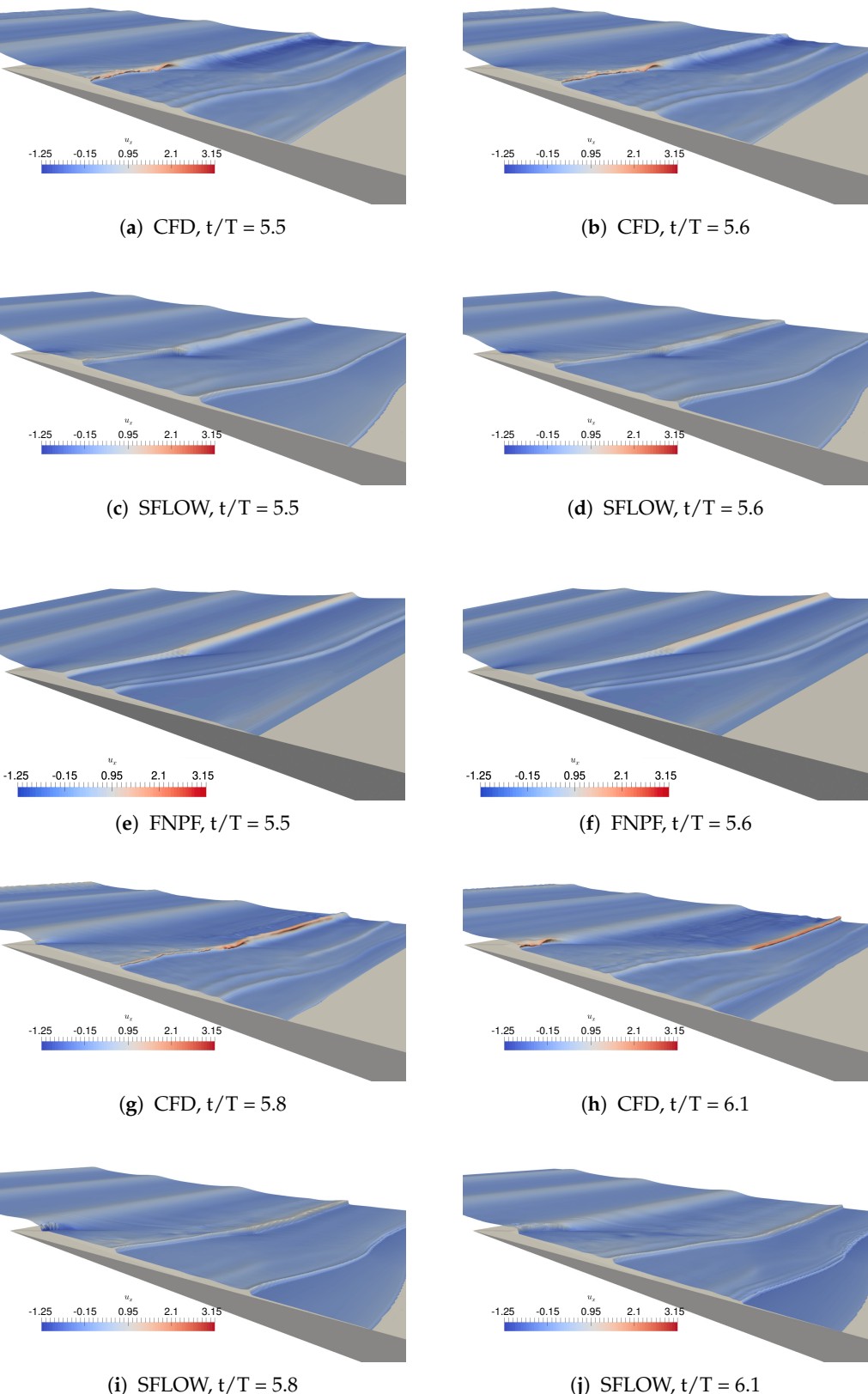

Figure 11. *Cont.*

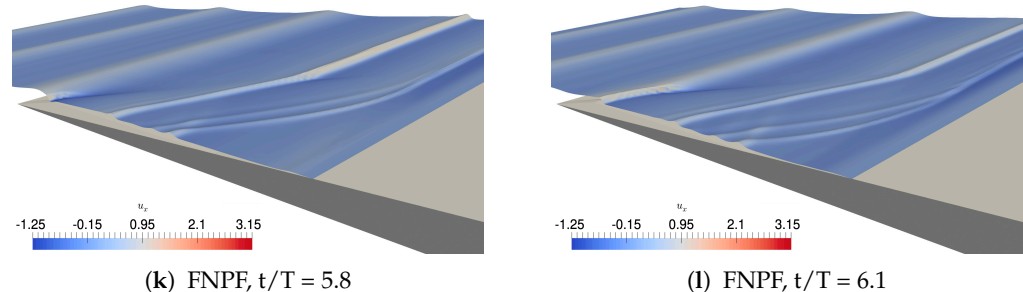

**(k)** FNPF, t/T = 5.8      **(l)** FNPF, t/T = 6.1

**Figure 11.** Free surface elevations with velocity contours at different time steps for three-dimensional wave breaking on a reef calculated using CFD and SFLOW.

The computational grid, computational resource and computational time from the three models are compared in Table 7. The computational speed gains from REEF3D::SFLOW and REEF3D::FNPF in a 3D simulation are seen to be even more prominent in comparison to the CFD solver, with a speedup factor of 60 and 800, respectively. On the other hand, the computational speed of REEF3D::CFD is compensated by the fact that REEF3D::CFD is the only model that is able to represent a correct geometry of an overturning breaker.

**Table 7.** Comparison of total number of cells $N_t$ and simulation time $T_s$ in seconds for the simulation of wave propagation over a submerged bar using the three modules.

| Module | $N_t$ | $T_s$ |
|---|---|---|
| REEF3D::CFD | 28,700,000 | 90 h |
| REEF3D::SFLOW | 450,000 | 5014.73 s |
| REEF3D::FNPF | 720,000 | 401.34 s |

## 4. Conclusions

In the presented manuscript, a comparative study of the three major types of phase-resolved wave models is presented with the use of the open-source hydrodynamics framework REEF3D. The development and numerical implementation of REEF3D are explained extensively to show the numerical consistency as well as differences among the wave models. The benchmark studies provide an insight into the strengths and limitations of each type of the wave modelling technique in terms of their computational performance as well as their limitations in different types of wave hydrodynamic phenomena. Thanks to the fact that all three models are implemented in the same numerical framework, an objective comparison is presented, which is not influenced by the various numerical implementations from different developers.

REEF3D::CFD solves the incompressible Navier–Stokes equations with a RANS turbulence model. Here, the pressure is solved on a staggered grid using the projection method. This ensures a tight pressure–velocity coupling. The model benefits from the utilization of a level set function to capture the motion of the free surface implicitly. In the numerical wave tank, the waves are generated and absorbed with either the relaxation method or using Dirichlet boundary conditions.

REEF3D::SFLOW reduces the computational costs significantly by solving the depth-averaged shallow water equations with a non-hydrostatic extension based on a quadratic vertical pressure profile. In comparison to existing approaches, like Boussinesq-type models or multi-layer approaches, the system of equations is solved with the projection method and high-order discretization schemes. This increases the stability of the computation through simpler terms in the equation and semi-implicit calculations for the pressure. Furthermore, the model benefits from the parallelization strategy in REEF3D which enables the simulation of large scale wave propagation near shores.

REEF3D::FNPF closes the gap between the efficient 2D shallow water solver and the accurate CFD solver for wave propagation problems as the FNPF potential flow solver is not restricted by water depth. By solving the three-dimensional Laplace equation with nonlinear boundary conditions for the free surface and the bottom, no simplifying assumptions regarding the wave characteristics or bottom slope are taken into account. At the same time, the use of a $\sigma$-coordinate system removes the additional cost of a two-phase approach. The model employs high-order discretization schemes in space and time which allows for larger cell sizes and time steps. Typically, ten cells in the vertical direction are sufficient to obtain accurate wave propagation. Very fast parallelized algorithms for solving the system matrix ensure the computational efficiency and enable the application for large-scale problems in deep and shallow water.

The performance of the presented modules has been tested and compared for several benchmark applications. The direct comparisons for regular waves show that all approaches are capable of predicting the wave propagation in their range of applicability. The challenging submerged bar case revealed very good accuracy of REEF3D::CFD and REF3D::FNPF, whereas the shallow water model fails due to its theoretical limitations. The two-dimensional wave breaking case shows that all three models are able to represent a correct wave energy dissipation during a breaking process. In the case of the three-dimensional wave breaking case, REEF3D::CFD and REEF3D::SFLOW capture the second breaking wave more accurately since both represent the swash zone dynamics better. The CFD based numerical wave tank is the only model that accurately represents the physics of wave propagation including complex overturning wave breaking. The computational speed gains from REEF3D::SFLOW and REF3D::FNPF in comparison to REEF3D::CFD are found to be by factors of about 10 and 40 on average for 2D simulations and 60 and 800 for the 3D simulation. The higher computational demands of the CFD model are compensated by that fact that it is the only model capable of representing the geometry of an overturning wave breaker accurately, which is important for studies on slamming load on structures.

With the strengths and limitations of each numerical models in mind, the future work will focus on the coupling of the different modules within REEF3D. A one-way coupling will use the propagated waves from a potential theory model as input waves in the CFD simulations. Two-way coupling processes will be interesting for applications in marine engineering with strong fluid–structure interactions.

**Author Contributions:** Conceptualization, H.B.; methodology, C.P.; software, H.B.; validation, W.W., A.K. and T.M.; formal analysis, W.W., A.K. and T.M.; investigation, W.W. and A.K.; writing–original draft preparation, W.W. and A.K.; writing–review and editing, H.B. and T.M.; visualization, W.W. and A.K.; supervision, H.B. All authors have read and agreed to the published version of the manuscript.

**Funding:** This research was funded by the Norwegian Public Roads Administration Grant No. 304624 and the research is in cooperation with the project funded by the Research Council of Norway Grant No. 267981.

**Acknowledgments:** The authors are grateful for the grants provided by the Research Council of Norway under the HAVBRUK2 project (No. 267981). This study has also been partially carried out within the E39 fjord crossing project (No. 304624) and the authors are grateful for the grants provided by the Norwegian Public Roads Administration. This research was supported with computational resources at NTNU provided by NOTUR (Norwegian Metacenter for Computational Sciences, http://www.notur.no) under project No. NN2620K.

**Conflicts of Interest:** The authors declare no conflict of interest. The funders had no role in the design of the study; in the collection, analyses, or interpretation of data; in the writing of the manuscript, or in the decision to publish the results.

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
