# Peer review of "A Comparison of Different Wave Modelling Techniques in An Open-Source Hydrodynamic Framework"

_jmse, doi:10.3390/jmse8070526_

Round 1
Reviewer 1 Report
The submitted manuscript discusses the different modes of use of REEF3D, an open source program for wave modelling in shallow water. I enjoyed very much reading this paper which is quite well written and is very clear in its exposition. Several example problems are considered which aid the reader in understanding the application of the model to a number of test cases, all of which are interesting from the point of view of offshore engineering. Both one dimensional and two dimensional cases are considered.
I have read this work and I am of the opinion that it will be of great value to engineers who seek to use REEF3D to solve problems in ocean engineering. The paper is quite stimulating from the perspective of important problems in the field and I highly recommend the paper for publication in the Journal of Marine Science and Engineering.
Author Response
The authors thank the reviewer for the acknowledgement of the work and wish the manuscript is of value to the audience
Reviewer 2 Report
The manuscript is about the REEF3D numerical model, being the performance of different modules validated and compared using several benmark cases. I recommend to include more validation cases, to be sure about the right working of the numerical model. The rest of validation cases can be shows in a summary way.
Author Response
The authors thanks the reviewer's interest in seeing more cases, the response is in the attached rebuttal.

Reviewer 3 Report
The article considers three wave numerical models within the framework of the REEF3D software package and conducts benchmarks on a series of various test problems in 2D and 3D.
The paper shows the strengths and limitations of each model. The conclusion also provides recommendations and justifications for choosing a model.
A detailed analysis is carried out for 2D dimensional problems: the grid convergence is checked, a comparison is made on several problems and with an experiment, where the results are shown to coincide. As for the 3D task, it has been performed somewhat limitedly; I would like to see a check for grid convergence and a comparison with the experiment. Despite this, this article will be useful to both users of the REEF3D software package and users of other software products built on the same sets of equations.
Despite this, a few remarks appeared:
- It is necessary to explain why the k-w turbulence model was used, not the more modern SST (Menter’s Shear Stress Transport[1]), which lacks the disadvantages of both k-w and k-ε models? Perhaps this could help to avoid problems with the damping of fluctuations near the free surface.
- It is required to supplement the description of the grids used to solve the three-dimensional problem. What type of mesh was used and how was it constructed? Why grid convergence was not investigated for a three-dimensional problem, perhaps then it was possible to overturning wave breaker using REEF3D::SFLOW or REEF3D::FNPF.
- It is required to explain why there is no comparison of 3D calculation and experiment, at a time when such a comparison is present for 2D. Since in view of the differences in the results obtained by different approaches, it is not clear which of them is more realistic and more priority in 3D.
- In figure 4 there is a comparison with the experiment, but nowhere experiment described and there are no references.
In general, this work is useful for those who perform calculations on REEF3D and deserve to be published.
[1] Menter, F. R. (August 1994). "Two-Equation Eddy-Viscosity Turbulence Models for Engineering Applications". AIAA Journal. 32 (8): 1598–1605. Bibcode:1994AIAAJ..32.1598M. doi:10.2514/3.12149.
Author Response
The authors thank the reviewer's comments, the detailed responses can be found in the attached rebuttal.

Round 2
Reviewer 2 Report
The authors have not reviewed the paper according to my comments
Author Response
The authors thank the reviewer for the suggestion. The previous rebuttal explained why the authors chose a small but representative cases to keep the article focused and concise. Extensive validations of each model can be found in the attached publications and thus not included in this manuscript:
REEF3D::CFD:
[1] Bihs H., Kamath A., Alagan Chella M., Aggarwal A. and Arntsen Ø. A., 2016, A new level set numerical wave tank with improved density interpolation for complex wave hydrodynamics, Computers& Fluids, https://doi.org/10.1016/j.compfluid.2016.09.012.
REEF3D::SFLOW:
[2] Wang W., Martin T., Kamath A. And Bihs H. 2020, An improved depth-averaged non-hydrostatic shallow water model with quadratic pressure approximation, International Journal For Numerical Methods In Fluids, http://dx.doi.org/10.1002/d.4807.
REEF3D::FNPF:
[3] Bihs H., Wang W., Martin T. And Kamath A. 2020. REEF3D::FNPF - A flexible fully nonlinear potential flow solver. Journal of Offshore Mechanics and Arctic Engineering, 142, https://doi.org/10.1115/1.4045915
